# Synthesis and Theoretical Studies of Aromatic Azaborines

**Pipsa Hirva [1], Petri Turhanen [2]  and Juri M. Timonen [2,3],***

[1] Department of Chemistry, Faculty of Natural Sciences and Forestry, University of Eastern Finland, P.O. Box 111, 80101 Joensuu, Finland; pipsa.hirva@uef.fi

[2] School of Pharmacy, Faculty of Health Sciences, Biocenter Kuopio, University of Eastern Finland, P.O. Box 1627, 70211 Kuopio, Finland; petri.turhanen@uef.fi

[3] Drug Research Program, Division of Pharmaceutical Chemistry and Technology, Faculty of Pharmacy, University of Helsinki, Viikinkaari 5E, P.O. Box 56, 00014 Helsinki, Finland

* Correspondence: juri.timonen@helsinki.fi

**Abstract:** Organoboron compounds are well known for their use as synthetic building blocks in several significant reactions, e.g., palladium-catalyzed Suzuki-Miyaura cross-coupling. As an element, boron is fascinating; as part of a molecule it structurally resembles a three-valent atom, but if there is a lone pair of electrons nearby, the boron atom's empty p-orbital may capture the lone pair and form a covalent bond. This is the main aspect that is challenging chemistry during the synthesis of boron containing molecules and may lead into unexpected reactions and products. To study this, we synthesized and studied novel aromatic azaborines for better understanding of their structures and reactions. Here, we report a one-pot method for the synthesis of substituted aromatic azaborines and computational studies of their structure to explain their observed chemical properties.

**Keywords:** azaborine; boron heterocycle; B-N heterocycle; Horner-Wadsworth-Emmons reaction; DFT; QTAIM; high performance counter current chromatography; HPCCC

## 1. Introduction

Boron containing heterocycles have met emerging interest in recent years. As an element, boron is a unique heteroatom which differs remarkably from the other heteroatoms because it has an empty p-orbital. This property drastically changes the nature of compounds bearing a boron atom and thus numerous applications have been developed for a wide range of boron containing heterocycles in the field of medicinal chemistry [1], materials sciences and catalysis [2] as well as in optoelectronics [3]. The most recent interest in boron containing heterocycles has been related to their fluorescence properties and possibilities to tune the fluorescence [4,5]. However, the development of novel materials, fluorescent compounds and boron-based drugs demands novel synthetic methodologies.

In medicinal chemistry, as well as in other target-oriented synthetic tasks, unexpected reactions are highly unwanted. In general, unexpected reactions follow, when a unique or a non-standard reaction is occurring. One challenging element is boron, in which an empty p-orbital makes it highly electron deficient and capable of withdrawing electrons which strongly affect the intra- and intermolecular properties and reactions of boron containing compounds.

According to the literature, only a few reports of aromatic azaborines (1-hydroxy-2,1-benzo[c]-1,2*H*-azaborine) (see Figure 1) have been published. Among the reported structures, only two are bearing a 3-carboxylate moiety. The first synthesis of these structures was presented by Choi and Lim in 2015, when they reported a mild, one-pot Horner-Wadsworth-Emmons olefination which led to an unexpected product, *N*-carboxybenzyl-1-hydroxy-6-methoxy-2,1benzo[c]-1,2*H*-azaborine-3-carboxylate instead of target indole [6]. Most recently, de Lescura et al. [7] published a study of microwave assisted synthesis of polycyclic azaborines from 2-formylphenylboronic acids. In

addition to these studies, only Saint-Louis et al. [3,5] and Gwynne et al. [8] have used 2-formylphenylboronic acid as a starting material for the synthesis of azaborines. A typical nitrogen source in the previous studies has been a lactam. Contrary to this, ethyl glycinate and 1-acetyl-3-indolinone were used by Lescura et al. and *N*-CBZ-phosphonyl glycine trimethyl ester by Choi et al.

**Figure 1.** Earlier and present research on synthesis of aromatic azaborines [3,5–7].

In our unpublished studies related to boronate amino acids, we have encountered unexpected reactions caused by the empty p-orbital of boron. One of these was the formation of azaborines despite attempts to protect boronic acid as an ester during Horner-Wadsworth-Emmons reaction. Inspired by these by-products, we report herein one-pot synthesis of some substituted methyl *1,2-H*-azaborine-3-carboxylates. Furthermore, we have studied the structural and electronic properties of the prepared compounds and their derivatives using density functional theory (DFT) methods. The main purpose of this work was to expand the knowledge of synthetic methodologies and understanding of the fascinating chemistry of 1-hydroxy-2,1benzo[c]-1,2*H*-azaborine-3-carboxylates.

## 2. Materials and Methods

All the chemicals were commercially available and were used as received, except if dry conditions were used, acetonitrile was dried before use by using standard methods. All reactions performed in dry conditions were carried out under a nitrogen atmosphere. Reactions were monitored using thin layer chromatography (TLC) or by $^1$H-NMR spectroscopy. All the NMR samples were dissolved in CDCl$_3$ and measured by Bruker Avance III HD

NMR spectrometer equipped with CryoProbe. [1]H and [13]C NMR spectra were recorded on a 600 MHz spectrometer operating at 600.2 and 150.9 MHz, respectively. Tetramehylsilane (TMS) was used as reference in NMR measurements. MS spectra were collected using high-resolution mass spectrometry (HRMS, Q Exactive Classic, Thermo Scientific, Bremen, Germany) equipped with a heated electrospray ionization (HESI) unit. Data were acquired in profile mode using data acquisition software Xcalibur 4.1.31.9 and MSTune 2.9 (Thermo Fisher Scientific, Breda, The Netherlands).

All target compounds (**1**, **4**, **5**, **6b**, **7**) were purified by high performance counter current chromatography (HPCCC) using HEMWat (hexane or heptane/ethyl acetate/methanol/water) solvent system 22. The purifications were performed in the normal phase mode of the HPCCC instrument (Spectrum, Dynamic Extractions, Slough, UK) and analytical (22.5 mL) column were used in all separations. The runs were monitored by UV-detection at 254 nm (there is the possibility to monitor four different wavelengths at the same time). As an example, HPCCC chromatogram of compound **1** purification can be found in the Supporting Information.

## 2.1. Molecular Modeling

All calculations were performed by applying Gaussian 09 software package [9]. The optimized geometries and simulated infrared spectra with no scaling for all molecular models of the structures were obtained by PBE0 functional [10] with 6-31G(d) basis set for all atoms. For the tests of the method and basis sets, see Supporting Information.

To study the electronic differences in the compounds, we performed topological charge density analysis with the QTAIM (Quantum Theory of Atoms in Molecules) [11] method, which allowed us to access the nature of the bonding via calculating different properties of the electron density at the bond critical points (BCPs). The analysis was performed with the AIMALL program [12] using the wavefunctions obtained from the single point DFT calculations of the models with optimized geometries.

## 2.2. Syntheses

General method for the synthesis of substituted methyl *N*-carboxybenzyl-1-hydroxy-2,1benzo[c]-1,2*H*-azaborine-3-carboxylates: In an oven dried flask, *N*-Cbz-*α*-phosphonoglycine trimethyl ester (1 equiv) was stirred with DBU (1, 2, or 3 equiv) in 5 mL of dry MeCN for 30 min under nitrogen. A mixture of 50 mg of boronic acid (1 or 2 equiv) dissolved into dry MeCN was added, and the reaction was continued for 3 h. The MeCN was removed in vacuo and the residue dissolved into 50 mL of DCM. The organic layer was washed once with 1M HCl (10 mL) and three times with water (10 mL). The organic layer was dried over MgSO$_4$ and evaporated into dryness. Crude products (112–147 mg) were purified by HPCCC.

## 2.3. HPCCC Purifications

We have earlier successfully used HPCCC for purification of bisphosphonate (medronate) monoesters [13] and biologically important adenosine triphosphate (ATP) analogues: AppppI [14] and ApppD [15]. Like in this reported research, those compounds were hard or even impossible to purify by conventional methods such as (flash) silica column chromatography or preparative TLC, besides HPCCC also offers a much higher capacity than HPLC and in our opinion, is better than HPLC in a number of other ways, as we have shortly discussed elsewhere. [PT1] In addition, according to our best knowledge, this is the first study where HPCCC has been used to purify any kind of boron containing organic molecules.

Methyl *N*-carboxybenzyl-1-hydroxy-2,1benzo[c]-1,2*H*-azaborine-3-carboxylate (1):
Yield 51 mg (46%), colorless semisolid. [1]H NMR (600 MHz, CDCl$_3$) δ 8.15 (d, *J* = 6.8 Hz, 1H, Ar), 7.79 (s, 1H B-OH), 7.60 (td, *J* = 7.5, 1.4 Hz, 1H, Ar), 7.48 (m, 2H), 7.39 (s, 5H, C$_6$H$_5$), 6.98 (s, 1H C=C*H*), 5.27 (s, 2H, C*H$_2$*C$_6$H$_5$), 3.56 (s, 3H, COOC*H$_3$*). [13]C-NMR (151 MHz, CDCl$_3$) δ 165.29 (1C), 156.99 (1C), 138.89 (1C), 134.11 (1C), 133.10 (1C), 132.35 (1C), 129.39

(1C), 128.94 (1C), 128.85 (2C), 128.75 (2C), 128.26 (1C), 128.07 (1C), 118.95 (1C), 69.68 (1C), 52.29 (1C). [11]B NMR (193 MHz, CDCl$_3$) δ 30.53 (s, 1B). HRMS: calcd. *m/z* for [C$_{18}$H$_{16}$BNO$_5$ + H]$^+$ 338.1200 found 338.1194.

Methyl *N*-carboxybenzyl-7-chloro-1-hydroxy-2,1benzo[c]-1,2*H*-azaborine-3-carboxylate (4):

Yield 30 mg (30%), colorless semisolid. [1]H-NMR (600 MHz, CDCl$_3$) δ 8.11 (d, *J* = 2.2 Hz, 1H), 7.76 (s, 1H), 7.55 (dd, *J* = 2.3, 8.3 Hz, 1H), 7.41 (d, *J* = 8.3 Hz, 1H), 7.40–7.37 (s, 5H), 6.93 (s, 1H), 5.28 (s, 2H), 3.56 (s, 3H). [13]C-NMR (151 MHz, CDCl$_3$) δ 165.03 (1C), 156.76 (1C), 137.16 (1C), 134.55 (1C), 133.96 (1C), 132.77 (1C), 132.62 (1C), 129.75 (1C), 129.43 (1C), 129.01 (1C), 128.88 (2C), 128.76 (2C), 117.75 (1C), 69.86 (1C), 52.34 (1C). [11]B NMR (193 MHz, CDCl$_3$) δ 30.20 (s, 1B). HRMS: calcd. *m/z* for [C$_{18}$H$_{15}$BClNO$_5$ + H]$^+$ 372.0810 found 372.0806

Methyl *N*-carboxybenzyl-1-hydroxy-6,7-dioxymethylene-2,1benzo[c]-1,2*H*-azaborine-3-carboxylate (5):

Yield 10 mg (10%), colorless semisolid. [1]H NMR (600 MHz, CDCl$_3$) δ 7.61 (s, 1H), 7.51 (s, 1H), 7.38 (bs, 5H), 6.89 (s, 1H), 6.88 (s, 1H), 6.05 (s, 2H), 5.26 (s, 2H), 3.56 (s, 3H). [13]C NMR (151 MHz, CDCl$_3$) δ 165.23 (1C), 157.05 (1C), 151.49 (1C), 148.39 (1C), 135.27 (1C), 134.15 (1C), 128.89 (1C), 128.81 (2C), 128.71 (2C), 128.16 (1C), 118.69 (1C), 111.27 (1C), 107.48 (1C), 101.45 (1C), 69.64 (1C), 52.20 (1C). [11]B NMR (193 MHz, CDCl$_3$) δ 30.37 (bs, 1B). HRMS: calcd. *m/z* for [C$_{19}$H$_{16}$BNO$_7$ + H]$^+$ 382.1098 found 382.1094.

Methyl 2-{[(benzyloxy)carbonyl]amino}-3-(2-borono-4-methoxyphenyl)prop-2-enoate (6b):

Yield 7 mg (7%). Colorless oil. [1]H NMR (600 MHz, CDCl$_3$) δ 8.03 (d, *J* = 1.5 Hz, 1H), 7.65 (s, 1H), 7.39 (s, 1H), 7.33 (bs, 5H), 6.85 (d, *J* = 8.7 Hz, 1H), 5.69 (s, 2H), 5.13 (s, 2H), 3.93 (s, 3H), 3.81 (s, 3H). [13]C NMR (151 MHz, CDCl$_3$) δ 165.97 (1C), 165.11 (1C), 139.42 (1C), 136.06 (1C), 134.46 (1C), 131.87 (1C), 128.77 (1C), 128.70 (1C), 128.47 (2C), 128.24 (2C), 128.17 (1C), 126.77 (1C), 110.19 (1C), 67.51 (1C), 55.71 (1C), 52.60 (1C).

Methyl *N*-carboxybenzyl-1-hydroxy-6-methoxy-2,1-benzo[c]-1,2*H*-azaborine-3-carboxylate (7):

Yield 22 mg (22%), colorless semisolid. [1]H-NMR (600 MHz, CDCl$_3$) δ 8.07 (d, *J* = 8.4 Hz, 1H), 7.67 (s, 1H), 7.39–7.38 (m, 5H), 7.04 (dd, *J* = 2.4, 8.4 Hz, 1H), 6.92 (d, *J* = 2.4 Hz, 1H), 6.91 (s, 1H), 5.27 (s, 2H), 3.88 (s, 3H), 3.55 (s, 3H). [13]C NMR (151 MHz, CDCl$_3$) δ 165.32 (1C), 162.86 (1C), 156.96 (1C), 140.89 (1C), 134.90 (1C), 134.14 (1C), 130.07 (1C), 128.92 (1C), 128.85 (2C), 128.74 (2C), 118.79 (1C), 115.76 (1C), 111.40 (1C), 69.62 (1C), 55.31 (1C), 52.28 (1C). HRMS: calcd. *m/z* for [C$_{19}$H$_{18}$BNO$_6$ + H]$^+$ 368.1305 found 368.1301

## 3. Results

### 3.1. Syntheses

To study the details of the reaction, we started to modify the reaction procedure and conditions reported in the literature for Horner-Wadsworth-Emmons reaction. At first, we studied four different kinds of organic bases: 1,8-diazabicyclo[5.4.9]undec-7-ene (DBU), di-isobutylethylamine (DIPEA), triethyl amine (TEA) and *N*-methyl morpholine (NMM) (Table 1, entries 1–4). Interestingly, the only base that led to product was DBU. In the next step, we studied the effect of solvent by comparing three different aprotic polar solvents: dichloromethane (DCM), tetrahydrofuran (THF) and acetonitrile (MeCN) (Table 1, entries 1, 5 and 6). The solvents were selected based on the literature, where similar reactions have been carried out in, e.g., DCM or THF. [6,16] During the studies, we noticed that the difference between MeCN and THF was insignificant and we elected to use more eco-friendly MeCN as a solvent in the further reactions. To optimize more of the reaction conditions, the reaction time was also monitored (Table 1, entries 6–11). According to the results, 3 h reaction time was the most suitable (Table 1 entry 9). The longer reaction time did not affect the conversion of the product.

To study the effect of stoichiometric ratio of starting materials and DBU, we carried out the reactions with 1 and 2 equiv of boronic acid and with 1, 2, or 3 equiv of DBU (entries 12–17). The best results were obtained when the excess of DBU was used in the reactions (2 or 3 equiv compared to phosphonoglysine, Table 1, entries 13 and 14). However, the use of excess of boronic acid did not have an effect at all (44% and 45% purified yields, respectively (Table 1 entries 15–17)).

**Table 1.** Optimization of the base for the synthesis of **1**.

| Entry | Base | Solvent | Time | Temperature | Equiv (Boronic Acid) | Equiv (Glynate) | Equiv (Base) | Yield/Conversion |
|-------|------|---------|------|-------------|----------------------|-----------------|--------------|------------------|
| 1 | DBU | DCM | 16 h | rt | 1.0 | 1.05 | 1.1 | 136 mg/25% [a] |
| 2 | TEA | DCM | 16 h | rt | 1.0 | 1.05 | 1.1 | n.d. [a] |
| 3 | DIPEA | DCM | 16 h | rt | 1.0 | 1.05 | 1.1 | n.d. [a] |
| 4 | NMM | DCM | 16 h | rt | 1.0 | 1.05 | 1.1 | n.d. [a] |
| 5 | DBU | THF | 16 h | rt | 1.0 | 1.05 | 1.1 | 102 mg/39% [a] |
| 6 | DBU | MeCN | 16 h | rt | 1.0 | 1.05 | 1.1 | 132 mg/31% [a] |
| 7 | DBU | MeCN | 8 h | rt | 1.0 | 1.05 | 1.1 | 32% [a, b] |
| 8 | DBU | MeCN | 4 h | rt | 1.0 | 1.05 | 1.1 | 31% [a, b] |
| 9 | DBU | MeCN | 3 h | rt | 1.0 | 1.05 | 1.1 | 32% [a, b] |
| 10 | DBU | MeCN | 2 h | rt | 1.0 | 1.05 | 1.1 | 5% [a, b] |
| 11 | DBU | MeCN | 1 h | rt | 1.0 | 1.05 | 1.1 | n.d. [a, b] |
| 12 | DBU | MeCN | 4 h | rt | 1.0 | 1.0 | 1.0 | 24% [c] |
| 13 | DBU | MeCN | 4 h | rt | 1.0 | 1.0 | 2.0 | 49% [c] |
| 14 | DBU | MeCN | 4 h | rt | 1.0 | 1.0 | 3.0 | 46% [c] |
| 15 | DBU | MeCN | 4 h | rt | 2.0 | 1.0 | 1.0 | 36% [c] |
| 16 | DBU | MeCN | 4 h | rt | 2.0 | 1.0 | 2.0 | 45% [c] |
| 17 | DBU | MeCN | 4 h | rt | 2.0 | 1.0 | 3.0 | 35% [c] |
| 18 | NaH + DBU | ACNe | 4 h | rt | 1.2 + 1.0 | 1.0 | 1.0 | 114 mg/29% [a] |

[a] Conversion measured by $^1$H NMR spectrum by comparing the product's COOMe and/or Ph-CH$_2$- signals with the phosphonoglysine's methyl and/or Ph-CH$_2$-signals. [b] Sample taken from the reaction mixture. [c] Purified yield.

The yield of products increased when two equivalents of DBU were used, and, therefore, we propose that it is possible that DBU forms a salt with boronic acid and thus prevents re-protonation of intermediate **Ib,** which is proposed to prevent the formation of the desired product (See the Discussion below). Park, Yoshino and Tomiyasu have studied the base assisted Horner-Wadsworth-Emmons reaction of 4-formylphenylboronic acid and *N*-(benzyloxycarbonyl)phosphonoglycine trimethyl ester with DBU, NaH and 1,1,3,3-tetramethylguanidine (TMG) as bases and showed also that the DBU was the most potent base for the reaction [16]. For the study of the effect of a combination of NaH and DBU, we activated phosphonoglycine with NaH followed the addition of the solution of 1eq of boronic acid and DBU. However, this did not increase the conversion of the desired product but rather increased observed byproducts (Table 1 entry 18). The optimized reaction conditions can be seen in Scheme 1.

**Scheme 1.** Synthesis of substituted methyl *N*-carboxybenzyl-1-hydroxy-2,1-benzo[c]-1,2*H*-azaborine-3-carboxylates. (a) 2-3 eq DBU, MeCN, rt, N$_2$, 30 min, (b) MeCN, rt, N$_2$, 3 h.

The optimized method (see Scheme 1) was then utilized to five aromatic 2-formylboronic acid derivatives to study the effect of nature of the boronic acid, which would yield the products presented in Table 2. In general, the method was not optimized systemically for each different boronic acids but rather the conditions were selected based on the results of the primary optimization and then slightly tuned by trying different stoichiometric ratios. Yields and best conditions are presented below in Table 2. Based on our results, it seems that substitution of the aromatic boronic acid greatly affects the formation of the products. Despite several attempts, the synthesis of **2**, **3**, and **6** failed. Interestingly, we found a small amount (7% purified yield) of a classical Horner-Wadsworth-Emmons product (**6b**, Table 2) instead of azaborine **6**. Furthermore, the replacement of formyl's hydrogen with more steric methyl group in **2** seems to block the reaction totally.

**Table 2.** Results of syntheses with different boronic acids.

| Boronic Acid | Yield | Best Conditions | Boronic Acid | Yield | Best Conditions |
|---|---|---|---|---|---|
| **2** | n.d. | | **5** | 10% | 100 mg (2 equiv) boronic acid, 86 mg (1 eq) trimethyl phosphonoglycinate, 116 µL (3 equiv) DBU. |
| **3** | n.d. | | **6b** | 7% | 50 mg (1 equiv) boronic acid, 85 mg (1 eq) trimethyl phosphonoglycinate, 116 µL (3 equiv) DBU. |
| **4** | 30% | 100 mg (2 equiv) boronic acid, 90 mg (1 eq) trimethyl phosphonoglycinate, 81 µL (2 equiv) DBU. | **7** | 22% | 100 mg (2 equiv) boronic acid, 92 mg (1 eq) phosphonoglycine trimethyl ester, 125 µL (3 equiv) DBU. |

In order to study the chemical reactions of CBZ-protected methyl azaborine carboxylates, we tested the normal protocols to remove CBZ and methyl ester protection groups (Scheme 2). Removal of methyl ester was studied using LiOH and NaOH solutions up to 50% (*m/w*) concentrations at temperatures up to 50 °C for up to 7 days. To remove CBZ group, 5 atm H$_2$ with 10 mol% Pd/C as catalyst was used with reaction times up to 72 h. Unexpectedly, these methods failed to deprotect compound **1**. Furthermore, the compound showed high stability by over 90% recovery of the starting material.

**Scheme 2.** Modifications of azaborine **1**. The additional aromatic stabilization of B-N ring is proposed to block reactivity of CBZ-N-C-COOMe side of the molecule.

The most probable explanation for the high stability is the aromatic-like nature of the six-membered B-N bond containing ring. Because boron is electron deficient, it withdraws the lone pair electrons of nitrogen causing the electron withdrawing effect which also affects the adjacent carbon and even further to the carbonyl carbon of methyl ester and makes the ester extremely stable against the hydrolysis. The same aromaticity affects the adjacent carbonyl carbon of CBZ as well, which is the most probable explanation for the stability against Pd (0) catalyzed hydrogenation.

*3.2. Molecular Stability*

In both $^1$H and $^{13}$C NMR spectra, all the compounds showed peaks for a distinct structure after long exposure in high vacuum, exposure to heating in NaOH solution or after exposure to different solvent systems after purification by High Performance Counter Current Chromatography (HPCCC). Major differences were found in the region of benzylic methylene protons that showed an AX doublet and in the region of methyl ester group. However, the structures had higher similarity at the aromatic region according to the $^1$H and $^{13}$C NMR spectra. This finding led us to consider the possibility of stable conformers giving these different chemical shifts. To study the conformational space of the CBZ and COOMe side of the molecule in **1**, we performed a 2D conformational analysis using DFT calculations. The rotatable bonds of the CBZ and COOMe, T1 and T2 (Figure S2), respectively, showed several local minima having hydrogen bond interactions between phenyl ring of CBZ and the carbonyl of the methyl ester. However, the energy difference between these local minima was 14 kJmol$^{-1}$ at the most, indicating that all of the conformations would be equally favorable at ambient temperatures. The conformers could explain the splitting of the NMR signal of the methylene protons. On the other hand, conformations with more closed geometry (1B-D, Figure S2) were found to stabilize the structures via hydrogen bonding, which can also explain the weaker reactivity against hydrolysis or removal of CBZ.

To establish more information on the relative stability, we performed a systematic computational investigation on the effect of the structural modifications on the properties of 1-hydroxy-2,1-benzo[c]-1,2*H*-azaborine-3-carboxylates. Full geometry optimization for the synthesized compounds **1–7** (Table 2) as well as some computationally modified derivatives of the key compound **1, 8–15** (Figure 2) was performed in vacuum at the selected DFT level of theory. Figures S3–S6 (Supporting Information) show the important optimized parameters for the compounds. There are quite small differences in the overall geometry of the compounds. Most of the compounds **1–15** showed essentially planar geometry of the B-N ring indicating substantial aromatic nature of the B-N bond. Only compounds **13** and **15** had a more twisted B-N ring structure and hence less aromaticity.

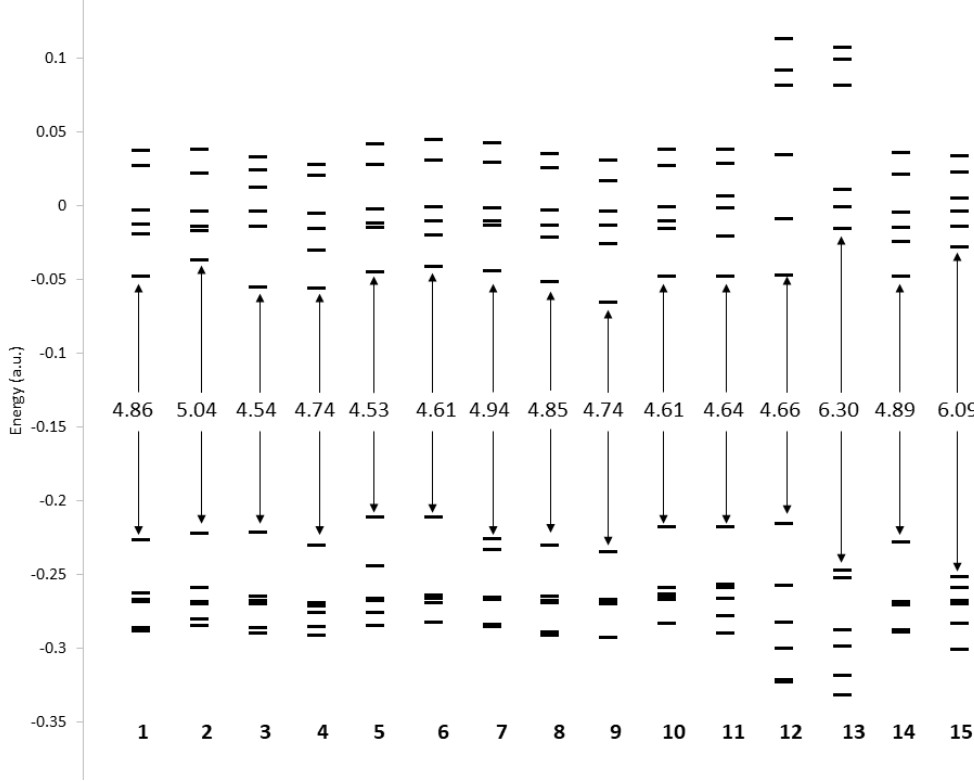

**Figure 2.** Structures of the computationally optimized azaborine-3-carboxylates.

The stability of the models was estimated by calculating the HOMO-LUMO gaps of all compounds. Figure 3 shows the FMO energy diagrams and the corresponding energy gaps (eV) for compounds **1**–**15**.

**Figure 3.** Frontier molecular orbital (FMO) energy diagrams (a.u.) and HOMO–LUMO gaps (eV) calculated at the DFT level of theory for compounds **1**–**15**.

In general, there are only slight differences in the HOMO-LUMO gaps of the molecules. Only the non-planar structures 13 and 15 show rather large stabilization of HOMO energy and destabilization of the LUMO energy, which leads to an increase in the HOMO-LUMO gap by over 1 eV.

*3.3. Intramolecular Interactions*

To investigate the effect of the different structural modifications on the nature of the bonding and the charge distribution in the aromatic azaborines, we conducted topological charge density analysis via QTAIM (Quantum Theory of Atoms in Molecules) methods. Table S1 lists selected properties of electron density at the bond critical points (BCPs) for the parent compound **1**. The numbering scheme of the BCPs is presented in Figure S7.

Since we were especially interested in the nature of the B=N bond, Table 3 compares the properties at the corresponding BCP for the selected set of compounds. The values for all studied compounds **1–15** can be seen in the Supporting Information (Table S2).

**Table 3.** Properties of the electron density at the B-N bond critical point (BCP #1, see Figure S7) of the selected aromatic azaborines. $\rho$ = electron density, $|V|/G$ = ratio between potential energy density and kinetic energy density, DI = delocalization index, $E_{BCP}$ = total energy at the BCP.

|  | **1** | **3** | **4** | **7** | **9** | **10** | **11** | **12** | **13** | **14** | **15** |
|---|---|---|---|---|---|---|---|---|---|---|---|
| $\rho$ | 1.19 | 1.17 | 1.20 | 1.18 | 1.20 | 1.20 | 1.24 | 1.31 | 1.33 | 1.19 | 1.19 |
| $|V|/G$ | 1.51 | 1.51 | 1.51 | 1.51 | 1.51 | 1.51 | 1.51 | 1.49 | 1.51 | 1.51 | 1.53 |
| DI | 0.41 | 0.41 | 0.41 | 0.40 | 0.41 | 0.41 | 0.42 | 0.44 | 0.44 | 0.41 | 0.40 |
| $E_{BCP}$ | −532 | −523 | −537 | −527 | −538 | −538 | −566 | −610 | −618 | −533 | −524 |

The largest electron density and delocalization index, as well as the strongest B=N bond are in compounds **12** and **13**. In both, the strong N-H bond invites larger electron density at the B=N BCP, which strengthens the bond and increases the bond index. Otherwise, there are no major differences in the key parameters.

During the 2D conformational analysis of the CBZ and COOMe substituents in the parent compound **1**, several lower energy conformations than the initial one was found, as already discussed. Four lowest energy conformations were further studied via QTAIM analysis to see what the effect of intramolecular weak interactions would be on the stability of the conformations. Table 4 lists the properties of the electron density at the selected BCPs. The numbering scheme of the BCPs can be seen in Figure 4.

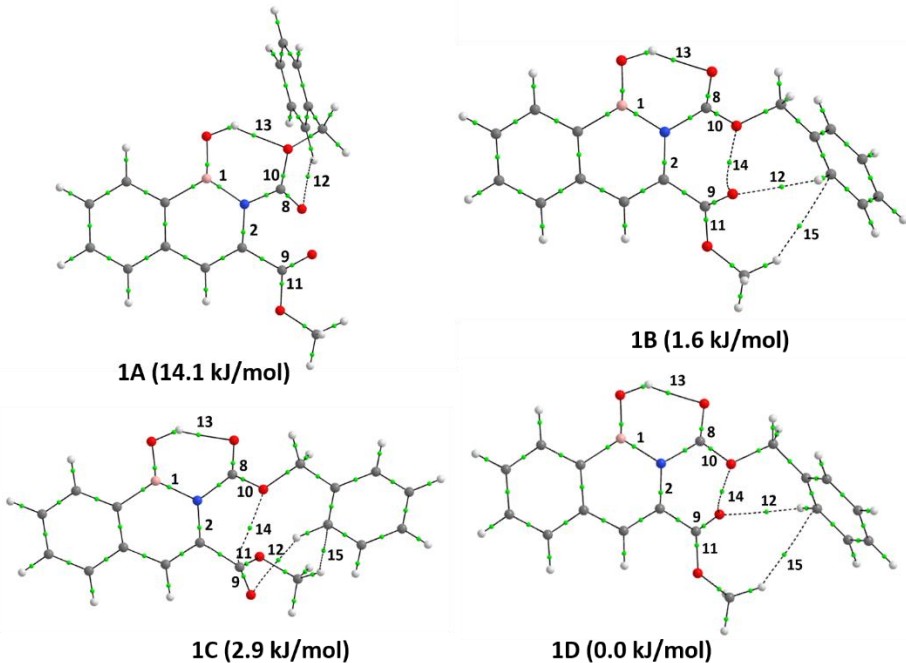

**1A (14.1 kJ/mol)**

**1B (1.6 kJ/mol)**

**1C (2.9 kJ/mol)**

**1D (0.0 kJ/mol)**

**Figure 4.** Bond paths and bond critical points (small green dots) in four lowest energy conformations of compound **1**.

**Table 4.** Properties of the electron density at the selected bond critical points (BCPs, see Figure 4) of the four lowest energy conformations of compound **1**. The parameters are d = bond distance, ρ = electron density, E = total energy at the BCP.

| BCP | Type | d [Å] | | | | ρ [e/Å³] | | | | E [kJmol⁻¹] | | | |
|---|---|---|---|---|---|---|---|---|---|---|---|---|---|
| | | 1A | 1B | 1C | 1D | 1A | 1B | 1C | 1D | 1A | 1B | 1C | 1D |
| 1 | B-N | 1.472 | 1.473 | 1.481 | 1.475 | 1.191 | 1.189 | 1.169 | 1.186 | −532 | −532 | −519 | −530 |
| 2 | N-C | 1.405 | 1.401 | 1.407 | 1.403 | 1.973 | 1.978 | 1.954 | 1.973 | −839 | −851 | −829 | −838 |
| 8 | C=O | 1.205 | 1.218 | 1.220 | 1.219 | 2.840 | 2.771 | 2.764 | 2.765 | −1973 | −1877 | −1864 | −1869 |
| 9 | C=O | 1.209 | 1.212 | 1.210 | 1.212 | 2.801 | 2.784 | 2.800 | 2.785 | −1940 | −1917 | −1933 | −1919 |
| 10 | C-O | 1.349 | 1.329 | 1.332 | 1.329 | 2.053 | 2.155 | 2.130 | 2.157 | −1104 | −1209 | −1194 | −1212 |
| 11 | C-O | 1.340 | 1.338 | 1.340 | 1.339 | 2.076 | 2.084 | 2.077 | 2.079 | −1158 | −1168 | −1158 | −1162 |
| 12 | O . . . H | 2.298 | 2.660 | 2.705 | 2.598 | 0.051 | 0.042 | 0.039 | 0.048 | −6 | −5 | −5 | −6 |
| 13 | O . . . H | 1.876 | 1.845 | 1.810 | 1.826 | 0.216 | 0.232 | 0.252 | 0.242 | −38 | −40 | −44 | −42 |
| 14 | O . . . O | - | 2.760 | 2.645 | 2.733 | - | 0.102 | 0.111 | 0.105 | - | −16 | −17 | −17 |
| 15 | H . . . C | - | 2.999 | 3.044 | 3.083 | - | 0.028 | 0.026 | 0.024 | - | −2 | −2 | −2 |

The properties of the electron density do not change much between the different conformations. The B-N bond is mostly affected only in the most "closed" conformation **1C**, which exhibits larger bond distance, less electron density and weaker interaction energy than the other three minimum energy conformations. The same applies to the C=O and C-O bonds of the CBZ and COOMe substituent, which show slightly modified properties of the electron density because of additional weak interactions.

## 4. Discussion

### 4.1. Proposed Reaction Mechanism

Because the boronic acids are Brønsted-acids, they may re-protonate the formed intermediate from glycine and thus prevent the reaction to proceed as already mentioned earlier. Among the studied organic bases, DBU was observed to be the only one that promotes the reaction, probably because it is the strongest base from the studied bases and also the only one with two nitrogen atoms in the structure. Both properties might be involved for the found activity.

Because we primarily found the formation of aromatic azaborines while studying the Horner-Wadsworth-Emmons reaction, we hypothesize the reaction mechanisms to be very similar (Scheme 3). The mechanism of Horner-Wadsworth-Emmons reaction is well-known in the literature and it consists of: (a) activation of phosphono moiety, (b) attack of carbonyl carbon, and finally (c) formation of four-membered ring with a consecutive elimination of dimethyl phosphate. In the literature, NaH is commonly used to form the activated intermediate **I**. In our studies, we found DBU to be the only organic base to yield the key compound **1** as described above. Our hypothesis is that other bases are not strong enough to avoid the re-protonation of the intermediate **I** by boronic acid. The finding that excess DBU increased the yield, supports the predicted reaction mechanism. In the literature, it has been proposed, that an electron withdrawing group at the α-position to phosphorus of the intermediate **III** enhances the reaction. In our study, the group was -NHCBZ, which is defined as an electron withdrawing group (EWG).

**Scheme 3.** The proposed reaction mechanism of the aromatic azaborine synthesis. (**a**) activation of phosphono moiety, (**b**) attack of carbonyl carbon, (**c**) formation of four-membered ring with a consecutive elimination of dimethyl phosphate.

However, the studies with different kinds of boronic acids illustrated that the major affecting factor is the substitution and nature of the boronic acid starting materials. In particular, difference between reactions of 5-methoxy-2-formylphenyl boronic acid and corresponding 4-methoxy isomer. When the methoxy is in *para* position to boronic acid moiety, it can donate electrons and assist the elimination of water while the methoxy group at the *meta* position is not able to promote this step (See Scheme 4).

**Scheme 4.** The assistance of the *para*-methoxy group that explains the different products from regioisomeric methoxy boronic acid starting materials.

## 4.2. Relative Stability and Nature of Bonding

DFT calculations on the synthesized compounds as well as structurally modified derivatives were performed to establish the relative stability of the molecules. As can be seen from Figure 3, the relative stability of most compounds is very similar. However, models **13** and **15** show larger HOMO-LUMO gaps indicating better stability. In both compounds, the HOMO is greatly stabilized and the LUMO destabilized as compared to the parent compound **1**. This can be attributed to the weaker aromaticity of the B-N ring, which leads to a more flexible structure and less ring stress.

In most of the compounds **1–15**, the FMOs look very similar than in **1** (Figure 5), both HOMO and LUMO are concentrated on the π system of the molecules and are more or less spread throughout the whole ring systems. On the contrary, in **13** and **15**, there are two nearly degenerate occupied orbitals, HOMO-1 and HOMO, which both are mainly expanded over the aromatic phenyl ring, and only slightly over the B-N ring, indicating less aromaticity of the latter. The larger concentration of electron density over the completely aromatic phenyl ring stabilizes the HOMO (and HOMO-1), therefore increasing the HOMO-LUMO gap. The difference in the electron distribution indicates larger stability of the compounds **13** and **15**.

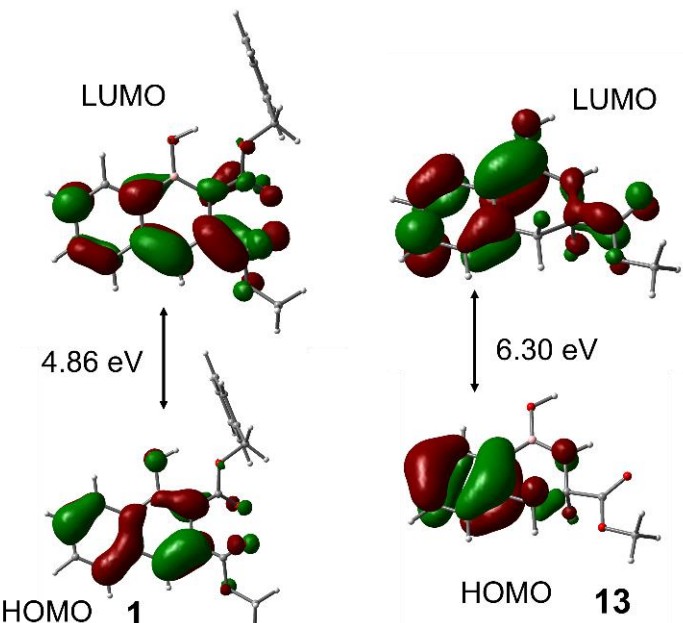

**Figure 5.** Comparison of the frontier molecular orbitals of compounds **1** and **13**.

Although the properties of the electron density do not change much between the different conformations, as obtained via the QTAIM analysis, the additional hydrogen-bonding and other weak interactions explain the strong stability of the compounds. The B-O-H⋯O=C(O)CBZ hydrogen bonding interaction (BCP #13, Figure 4) is especially strong and is therefore able to fix the torsion to a planar one. In the lowest energy conformations, the additional O⋯O interactions also limit the rotation of both carboxylate groups. The experimentally observed failure in formation of compounds **8** and **12** can be attributed to these weak interactions, which stabilize the more "closed" conformation and protect the reaction site effectively.

## 5. Conclusions

Even though medicinal chemistry very often aims towards a single specific target compound, the synthesis route may lead into an interesting side-guest. Aromatic azaborines have been synthetized at least once before as unexpected products. They have also been synthetized and studied intentionally and found to posess several interesting properties. Here, we have reported a one-pot synthesis method of four aromatic azaborines (**1**, **4**, **5**, and **7**), from which three are novel (**1**, **4**, and **5**), and studied their reactivity and structural properties. Most of the earlier published aromatic azaborines have contained more fused rings than the products we have reported here, which makes them much easier to separate and purify. In this light, the use of High Performance Counter Current Chromatography made it possible to separate and purify these compounds from the reaction mixture. Furthermore, the study shed light onto the high stability of the compound **1** towards deprotection of CBZ and COOMe in standard or even rough reaction conditions. However, there are

still questions unanswered, and the potency of boron containing heterocycles needs to be further studied.

**Supplementary Materials:** The following Supporting Information can be downloaded at: https://www.mdpi.com/article/10.3390/org3030016/s1, Figure S1: The parent compound **1**; Figure S2: 2D conformational analysis of **1**; Figures S3–S6: optimized structures of **1**–**15**; Figure S7: numbering of the BCPs in **1**; Figure S8: An example of HPCCC chromatogram for key compound **1**; Figures S9–S20: $^1$H, $^{13}$C and $^{11}$B NMR spectra of Key Compounds **1**, **4**, **5** and **7**; Table S1: properties of the electron density at selected BCPs of **1**; Table S2: properties of the electron density at the B-N BCP for **1**–**15**.

**Author Contributions:** Investigation, P.H., P.T. and J.M.T.; writing—original draft preparation, P.H. and J.M.T.; writing—review and editing, P.H., P.T. and J.M.T.; project administration, J.M.T. All authors have read and agreed to the published version of the manuscript.

**Funding:** This research received no external funding.

**Institutional Review Board Statement:** Not applicable.

**Informed Consent Statement:** Not applicable.

**Data Availability Statement:** Not applicable.

**Acknowledgments:** The authors would like to thank the School of Pharmacy, University of Eastern Finland for providing funding for the experimental work. Marko Lehtonen is acknowledged for providing the mass spectrometric data. Furthermore, Landry Amamea is acknowledged for his work leading to the finding of this method during his thesis. The authors would like to thank Maritta Salminkoski (retired) for her invaluable work in the synthesis laboratory. P.H. would like to acknowledge grants of computer capacity from the Finnish Grid and Cloud Infrastructure (persistent identifier urn:nbn:fi:research-infras-2016072533). J.M.T. would like to acknowledge The Finnish Cultural Foundation for a personal grant that made this study possible as a part of the LAT1 targeted BNCT research.

**Conflicts of Interest:** The authors declare no conflict of interest.

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
