# Peer review of "Synthesis and Theoretical Studies of Aromatic Azaborines"

_organics, doi:10.3390/org3030016_

Round 1
Reviewer 1 Report
This manuscript reports the synthesis of substituted methyl 1,2-H-azaborine-3-carboxylates, and their structural and electronic properties by using density functional theory (DFT) methods to understand their new chemistry. The obtained results are somewhat interesting to organic chemists. It is recommended for publication in Organics after revision subjected to the following points.
(1) The abstract can be shortened to focus on describing the work which have been done, and the obtained results.
(2) Lines 55-56: “In our previous studies related to boronate amino acids, we have encountered unexpected reactions caused by the empty p-orbital of boron.” The literature reported by authors should be cited.
(3) The potential application of the prepared compounds should be described.
Author Response
This manuscript reports the synthesis of substituted methyl 1,2-H-azaborine-3-carboxylates, and their structural and electronic properties by using density functional theory (DFT) methods to understand their new chemistry. The obtained results are somewhat interesting to organic chemists. It is recommended for publication in Organics after revision subjected to the following points.
- The abstract can be shortened to focus on describing the work which have been done, and the obtained results.
Reply: The abstract has been shortened.
- Lines 55-56: “In our previous studies related to boronate amino acids, we have encountered unexpected reactions caused by the empty p-orbital of boron.” The literature reported by authors should be cited.
Reply: These studies has not been published yet, so changed to “In our unpublished studies related”
- The potential application of the prepared compounds should be described.
Reply: There are currently ongoing studies in several fields of materials chemistry pursuing to develop novel molecular structures for novel materials for e.g. optoelectronics. The properties of boron containing heterocycles makes them potential for different applications requiring defined electronic properties. For example, organic LED (OLEDs) have been utilized in printable electronics. Also, the tunable fluorescence properties shown with some polycyclic azaborines has been speculated to be usable as chemosensors. However, our manuscript concentrates on studying and explaining properties of these compounds and thus forecasting applications for these compounds without any experimental studies related to such uses would be highly speculative. In the introduction we have explained with references the emerging interest towards boron containing molecules. We hope that these references and revision of the literature gives an idea for the reader of the field which this study might serve.
Reviewer 2 Report
The subject of this research are the aromatic azaborines. The authors aimed to expand the knowledge of synthetic methodologies and understanding of the chemistry of 1-hydroxy-2,1-benzo[c]-1,2H-azaborine-3-carboxylates. In fact, the research team synthetized and studied novel aromatic azaborines. They reported a one-pot method for the synthesis of substituted aromatic azaborines, and explained the observed chemical properties of the synthesized aromatic azoborines via computational studies.
То achieve the research goal, the team used modern analytical techniques, e.g. preparative high performance counter current chromatography with UV detection; 600 MHz NMR spectrometer equipped with CryoProbe; HESI-MS spectrometer; topological charge density analysis with the QTAIM-method.
The authors applied supplemental materials like 1H, 13C and 11B NMR spectra, HPCCC chromatograms and data from the topological charge density analysis with the QTAIM-method. These are raw data, which prove the truth, and indisputability of the experiment, and the right interpretation of the results obtained.
The paper is written well and concisely. The text is clear and easy to read. The conclusions consistent with the evidence and arguments presented, and they address the main question posed. The references cited are relevant, and the most of them are from the last 5-10 years.
I don’t have any questions and suggestions. I recommend accepting the article for publishing in its current form.
Author Response
Reply: Thank you for your kind and encouraging comments.
Reviewer 3 Report
In this manuscript the authors have described a one-pot method to synthesize aromatic azaborines and a theoretical study has been made to better understanding the structural characteristics as well as the chemical properties of this class of boron compounds. However, by concerning the applicability and the usefulness of this synthetic approach it has proved to be very limited since only four examples were isolated and either in very poor yield. On the other hand, in reference of the theoretical studies, it represents a very interesting research material that could be very useful for further studies in boron chemistry and especially for borines their applications.
- Abstract – line 17 - change “synthetized” by “synthesized”;
- Introduction – line 44 – In “N-carboxybenzyl-1-hydroxy-6-methoxy-2,1benzo[c]-1,2H-azaborine-3-carboxylate” the “N” must be in italic format. This aspect must be revised for all cited compound names;
- Figure 1 – the schemes are poorly made without any patterning. The reaction parameters described above and below of the reaction arrow must be formatted to follow a standard. The amounts of the reagents must be described. Change “RT” by “rt”; “160C” by “160 ºC”; “48h” by “48 h”; change “ACN" by “MeCN” not only in figure but throughout the manuscript.
- Change “eq” by “equiv” throughout the manuscript.
- Table 1 – Why did not test the influence of the temperature for the reaction behavior?
- I suggest repeating the experiment of the entry 18, but using 2 equiv of DBU;
- The reaction equation must be added at the top of the table; change “RT” by “rt”;
- On the foot note, the authors mentioned “conversion”. At last, the data represented in the table are yield or conversion?
- Page 5 – line 191 – the reference [16] is cited after the punctuation mark.
- Figure 2 – I recommend removing this figure and transferring the structures to the table 2.
Author Response
In this manuscript the authors have described a one-pot method to synthesize aromatic azaborines and a theoretical study has been made to better understanding the structural characteristics as well as the chemical properties of this class of boron compounds. However, by concerning the applicability and the usefulness of this synthetic approach it has proved to be very limited since only four examples were isolated and either in very poor yield. On the other hand, in reference of the theoretical studies, it represents a very interesting research material that could be very useful for further studies in boron chemistry and especially for borines their applications.
Reply: Thank you for your valuable comments on our manuscript.
- Abstract – line 17 - change “synthetized” by “synthesized”;
Reply: Has been corrected.
- Introduction – line 44 – In “N-carboxybenzyl-1-hydroxy-6-methoxy-2,1benzo[c]-1,2H-azaborine-3-carboxylate” the “N” must be in italic format. This aspect must be revised for all cited compound names;
Reply: All of the names corrected.
- Figure 1 – the schemes are poorly made without any patterning. The reaction parameters described above and below of the reaction arrow must be formatted to follow a standard. The amounts of the reagents must be described. Change “RT” by “rt”; “160C” by “160 ºC”; “48h” by “48 h”; change “ACN" by “MeCN” not only in figure but throughout the manuscript.
Reply: Done
- Change “eq” by “equiv” throughout the manuscript.
Reply: Done.
- Table 1 – Why did not test the influence of the temperature for the reaction behavior?
Reply: Our aim in this work was to study a mild method to prepare the key compounds. Based on our previous experience the elevated temperatures tend to increase intermolecular side reactions of boronic acids than enhance formation of B-N rings.
- I suggest repeating the experiment of the entry 18, but using 2 equiv of DBU;
Reply; Because the main result from using NaH as an activating base was increase of side-reactions (seen in the NMR as numerous new byproducts at the aliphatic region) we believe that carrying out the reaction with larger amount of organic base won’t actually enhance the reaction. Use of lowered temperature when using NaH would be necessary. However, we wished to prepare the compounds using a simple and easy-to-use one-pot method. The main purpose in this manuscript is to study the chemistry and properties of these compounds and the fine tuning of the reaction would be highly interesting topic for a deeper study concentrating purely on the synthesis.
- The reaction equation must be added at the top of the table; change “RT” by “rt”;
Reply: Done
- On the foot note, the authors mentioned “conversion”. At last, the data represented in the table are yield or conversion?
Reply: Edited the column’s caption to Yield/Conversion. The conversions are measured in the preliminary studies, in which the purification was not necessary because even the conversions were very low. Purification was used only when the conversion was high enough to give a useful yield.
- Page 5 – line 191 – the reference [16] is cited after the punctuation mark.
Reply: Corrected
- Figure 2 – I recommend removing this figure and transferring the structures to the table 2.
Reply: Done
Reviewer 4 Report
The manuscript by Pipsa Hirva, Petri Turhanen and Juri Timonen on “Synthesis and theoretical studies of aromatic azaborines” (#organics-1684602) has been reviewed. The work describes the one-pot synthesis and computational studies of novel and interesting aromatic azaborines. The paper describes also a convenient separation and purification method for the compounds, and introduce an interesting deprotection method of CBZ and COOMe groups that can be utilized also when preparing more complex molecules. The figures are of good quality, there is a good number of citations mentioned and the manuscript has no grammatical errors and typos. The paper can in my opinion be published in the journal Organics without additions.
Author Response

(The authors gave the same response as above.)

Round 2
Reviewer 3 Report
After to analyse the present form of the manuscript I consider It is suitable for publication on Organics since most of the reviewer appointments were considered and promptly adjusted by the authors.